# Linear Quadratic Optimal Control with the Finite State for Suspension System



**Qidi Fu** [1] , **Jianwei Wu** [2] , **Chuanyun Yu** [1], **Tao Feng** [1], **Ning Zhang** [1] and **Jianrun Zhang** [1,*]

1 School of Mechanical Engineering, Southeast University, Nanjing 211189, China
2 Institute of New Technology, Guangxi Liugong Machinery Co., Ltd., Liuzhou 545007, China
* Correspondence: zhangjr@seu.edu.cn

**Abstract:** The control algorithm could greatly help the suspension system improve the comprehensive performance of the vehicle. Existing control methods need to obtain the intermediate states, which are difficult to obtain directly or accurately when estimated by filters or observers. Thus, this paper proposed a new practical finite state LQR control method to deal with this problem. By combining with the output state of the finite sensor of the vehicle suspension system and weakening the unknown state as the goal, an optimization model is established with the design variables as the LQR weight coefficients. Then, the direct relationship between the current control input and the finite sensor output is obtained, and the finite state LQR control is realized. Taking the quarter-car suspension model as an example, the corresponding noise is added considering sensor accuracy, and the control performance of the four control methods is studied considering the uncertainties of suspension system parameters. In addition, the acceleration of sprung mass and the dynamic travel coefficient of suspension have been separately calculated by methods of finite state LQR control, LQR control, and PID control. The results show that there is not much difference between them under shock excitation or random excitation. However, the finite state LQR control method has the best comprehensive control performance in that its dynamic tire load coefficient is better than other methods; it could take into account the suspension work stroke coefficient, dynamic tire load coefficient, and sprung mass' acceleration of the vehicle suspension system at the same time. In order to realize the optimal control effect with limited sensor arrangement, the finite state LQR control method only needs to obtain the current sensor output and the current control input, without estimating the unknown intermediate state. By this means, the proposed control method greatly simplifies the design of the control system and has great advantages on practical value.

**Keywords:** suspension system; active control; finite state linear quadratic regulator; LQR; PID

## 1. Introduction

With the development of modern vehicle electronic control technology, active and semi-active suspensions are widely used in vehicles as they have incomparable advantages over traditional passive suspension. Active and semi-active suspensions can adjust the system parameters in real time according to road conditions and vehicle conditions to get better vibration reduction performance. Therefore, the development of a simple, efficient, and adaptable algorithm has become the key of active or semi-active suspension design. In recent decades, to obtain better suspension control performance, researchers have investigated and proposed a variety of suspension control algorithms, such as linear quadratic optimal control, skyhook control, sliding mode control, fuzzy logic control, neural network control, adaptive control, H∞ control, and so on [1–7].

However, to obtain favorable control performance, existing suspension control methods all desire excessive state information. Due to the availability of state information as well as the installation and cost of sensors, there is a contradiction between the superiority of state feedback in performance and the difficulty in physical implementation [8].

Aimed at the problem, a common solution is to design an observer based on the vehicle dynamics model. By utilizing system inputs and vehicle states that are easy to be measured, observers can reconstruct or identify unavailable states and unmeasurable parameters. To improve the accuracy of state estimation, make full use of sensor information, and reduce the cost of control system, various observer algorithms have been proposed recently, such as the Kalman filter [9,10], unscented Kalman filter [11–14], Luenberger observer [15–17], Sliding mode observer [18,19], etc. However, these observers are usually separated from the controller. Consequently, such observers not only increase the calculation burden of the control system but also increase the control parameters of the active suspension system, which makes it difficult to achieve the ideal control effect in practice.

Vehicle active/semi-active suspension is a complicated system, involving a control algorithm, parameter optimization, dynamic modeling, state estimation, system identification, signal processing, and other techniques. It is difficult to ensure desired vehicle control performance and reliability by purely focusing on control and estimation algorithms. Many advanced control methods can consider factors such as parameter uncertainty, system nonlinearity, and unmodeling. However, these methods all result in a complex control law and excessive adjustable parameters, which makes the application difficult and unreliable. In fact, to achieve the final control objectives, it is not necessary to get all the states of the system, but to use the existing sensor states to obtain the control output. Hence, to improve the practicality and reliability of the control system, the development of a control algorithm that makes full use of existing sensor information is desired. Combining the control theory, finite sensor arrangement, and advanced optimization, the effect of unknown states can be weakened, leading to the elimination of state errors, reduction of suspension active control parameters, and robustness against suspension uncertainties.

This paper combined linear quadratic regulator (LQR), finite sensor arrangement, and the modern control theory together and proposed finite state LQR (FSLQR) control. FSLQR weakens the effect of unknown states through optimization of LQR weight coefficients. Furthermore, the performance of FSLQR is studied through examples under different conditions where sensor noises and suspension uncertainties are considered.

## 2. Suspension Control Model

### 2.1. Quarter-Car Model

Figure 1 shows a quarter-car model of a suspension system. The motions of sprung and unsprung mass can be formulated as:

$$
\begin{aligned}
m_s \ddot{z}_s(t) &= -c_s\left(\dot{z}_s(t) - \dot{z}_u(t)\right) - k_s(z_s(t) - z_u(t)) + u(t) \\
m_u \ddot{z}_u(t) &= c_s\left(\dot{z}_s(t) - \dot{z}_u(t)\right) + k_s(z_s(t) - z_u(t)) - \\
&\quad c_t\left(\dot{z}_u(t) - \dot{z}_r(t)\right) - k_t(z_u(t) - z_r(t)) - u(t).
\end{aligned}
\tag{1}
$$

where $m_u$ and $m_s$ denote the unsprung mass and sprung mass, respectively. $z_r$, $z_u$ and $z_s$ are the vertical displacement of road surface, unsprung mass, and sprung mass. $k_t$ and $c_t$ are the stiffness and damping of tire, respectively. $u$ denotes the control input.

With state vector $\boldsymbol{x}(t) = \begin{bmatrix} x_1, & x_2, & x_3, & x_4 \end{bmatrix}^T = \begin{bmatrix} z_s - z_u, & z_u - z_r, & \dot{z}_s, & \dot{z}_u \end{bmatrix}^T$ and disturbance $\omega(t) = \dot{z}_r(t)$, Equation (1) yields the following suspension model [20]:

$$
\dot{\boldsymbol{x}}(t) = \boldsymbol{A}\boldsymbol{x}(t) + \boldsymbol{B}u(t) + \boldsymbol{E}\omega(t)
\tag{2}
$$

with matrices

$$
\boldsymbol{A} = \begin{bmatrix}
0 & 0 & 1 & -1 \\
0 & 0 & 0 & 1 \\
-\dfrac{k_s}{m_s} & 0 & -\dfrac{c_s}{m_s} & \dfrac{c_s}{m_s} \\
\dfrac{k_s}{m_u} & -\dfrac{k_t}{m_u} & \dfrac{c_s}{m_u} & -\dfrac{c_s + c_t}{m_u}
\end{bmatrix},
$$

$$
\boldsymbol{B} = \begin{bmatrix} 0, & 0, & \dfrac{1}{m_s}, & \dfrac{-1}{m_u} \end{bmatrix}^T
$$

$$
\boldsymbol{E}(t) = \begin{bmatrix} 0, & -1, & 0, & \dfrac{c_t}{m_u} \end{bmatrix}^T.
\tag{3}
$$

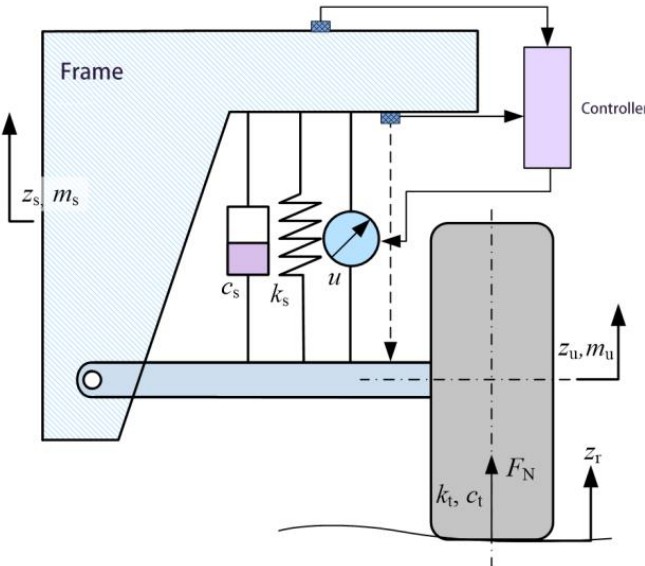

**Figure 1.** Quarter car suspension model.

### 2.2. Suspension Performance Index

The design of controlled vehicle suspension system aims to enhance the vehicle performance with regard to ride comfort and road holding. Dynamic tire load, work stroke, and acceleration of sprung mass are usually used to evaluate suspension performance. Given their magnitudes, the suspension performance index is defined as $\boldsymbol{y}_r(t) = \begin{bmatrix} y_{r1}(t) & y_{r2}(t) & y_{r3}(t) \end{bmatrix}^T$. $y_{r1}(t)$, $y_{r2}(t)$, and $y_{r3}(t)$ are the suspension work stroke coefficient (SWSc), dynamic tire load coefficient (DTLc), and acceleration of sprung mass, respectively. The performance index can be formulated as:

$$\boldsymbol{y}_r(t) = \boldsymbol{C}_r \boldsymbol{x}(t) + \boldsymbol{D}_r u(t) \tag{4}$$

where

$$\boldsymbol{y}_r = \begin{bmatrix} y_{r1}(t) \\ y_{r2}(t) \\ y_{r3}(t) \end{bmatrix} = \begin{bmatrix} (z_s - z_u) \cdot \frac{1}{z_{max}} \\ (z_u - z_r) \cdot \frac{k_t}{(m_s + m_u)g} \\ \ddot{z}_s \end{bmatrix}$$

$$\boldsymbol{C}_r = \begin{bmatrix} \frac{1}{z_{max}} & 0 & 0 & 0 \\ 0 & \frac{k_t}{(m_s + m_u)g} & 0 & 0 \\ -\frac{k_s}{m_s} & 0 & -\frac{c_s}{m_s} & \frac{c_s}{m_s} \end{bmatrix} \tag{5}$$

$$\boldsymbol{D}_r = \begin{bmatrix} 0 & 0 & \frac{1}{m_s} \end{bmatrix}^T$$

where $z_{max}$ represents the limited travel of suspension and $g$ denotes acceleration of gravity.

### 2.3. Ideal Sensor Output

The active suspension system commonly includes displacement sensors and acceleration sensors, usually arranged as shown in Figure 1. The ideal output of sensors that ignores noises can be obtained as:

$$\boldsymbol{y}_s(t) = \boldsymbol{C}_s \boldsymbol{x}(t) + \boldsymbol{D}_s u(t) \tag{6}$$

with matrices

$$\boldsymbol{C}_s = \begin{bmatrix} 1 & 0 & 0 & 0 \\ -\frac{k_s}{m_s} & 0 & -\frac{c_s}{m_s} & \frac{c_s}{m_s} \end{bmatrix} \text{ and } \boldsymbol{D}_s = \begin{bmatrix} 0 \\ \frac{1}{m_s} \end{bmatrix} \tag{7}$$

*2.4. State-Space Equation*

Considering the performance indexes of the suspension system, we can obtain the control-oriented state-space equation as shown of Equation (8). Generally, the control input and disturbance input are vectors containing multiple elements:

$$\begin{cases} \dot{x}(t) = Ax(t) + Bu(t) + E\omega(t) \\ y_s(t) = C_s x(t) + D_s u(t) \\ y_r(t) = C_r x(t) + D_r u(t) \end{cases} \tag{8}$$

Considering the uncertainties of actual suspension parameters and noises of sensors, the actual state-space equation can be given by:

$$\begin{cases} \dot{x}(t) = A(t)x(t) + B(t)u(t) + E\omega(t) \\ y_s(t) = C_s(t)x(t) + D_s(t)u(t) + v(t) \end{cases} \tag{9}$$

where, $\bullet(t)$ represents a time-varying matrix and $v(t)$ denotes the output noise.

## 3. Finite State LQR Control

To avoid the estimation of uncertain state variables, it is desired that the states measured by existing sensors only are enough to guarantee the control objectives. Aimed at the problem, FSLQR control method is proposed by optimizing the weight coefficients of LQR control.

*3.1. Linear Quadratic Regulator*

Quadratic performance function is established as Equation (10), based on the suspension performance index and control input:

$$\begin{aligned} J \quad &= \int_0^{+\infty} y_r^T(t) Q_r y_r(t) dt + \int_0^{+\infty} q_4 u^2(t) dt \\ &= \int_0^{+\infty} \left\{ \begin{array}{l} x^T(t) C_r^T Q_r C_r x(t) + 2x^T(t) C_r^T Q_r D_r u(t) + \\ u^T(t) \left( D_r^T Q_r D_r + q_4 \right) u(t) \end{array} \right\} dt \end{aligned} \tag{10}$$

The performance function can be further simplified as:

$$J = \int_0^{+\infty} \left\{ x^T(t) Q x(t) + 2x^T(t) N u + u^T R u \right\} dt \tag{11}$$

where $Q = C_r^T Q_r C_r$, $N = C_r^T Q_r D_r$, $R = D_r^T Q_r D_r + q_4$, and $Q_r = \text{diag}(q_1, q_2, q_3)$. $q_1$, $q_2$, $q_3$, and $q_4$ denote the weight coefficients of $y_{r1}(t)$, $y_{r2}(t)$, $y_{r3}(t)$, and $u(t)$, respectively. According to Riccati equation, the optimal feedback gain is given by:

$$K = R^{-1} \left( B^T P + N^T \right) \tag{12}$$

where $P$ is the solution of following Riccati equation:

$$A^T P + PA - (PB + N) R^{-1} \left( B^T P + N^T \right) + Q = 0 \tag{13}$$

Thus, the optimal control feedback is:

$$u_{\text{opt}}(t) = -Kx(t) \tag{14}$$

Considering the quarter-car model, $u_{\text{opt}}(t) = -(K_1 x_1(t) + K_2 x_2(t) + K_3 x_3(t) + K_4 x_4(t))$. However, only the acceleration of sprung mass $\ddot{z}_s$ (or $\dot{x}_3$) and suspension work stroke $x_1$ can be measured conveniently by vehicular sensors. Meanwhile, the velocity of suspension work stroke $\dot{x}_1$ can be obtained by the first derivative of $x_1$. Therefore, to achieve FSLQR control by only adopting the states from existing sensors, the feedback gain of tire dynamic deflection $K_2$ is required to be 0. Furthermore, $K_3 = -K_4$ is desired, as the absolute velocities of unsprung mass and sprung mass are very difficult to get. Based on this condition, it could be easier for us to realize the LQR control by getting the relative acceleration. By the proper design of optimization model, the LQR weight coefficients are optimized, and the desired LQR feedback law can be achieved.

### 3.2. Optimization Model of Finite LQR Control

The design variables are selected as:

$$\boldsymbol{X} = \begin{bmatrix} q_1 & q_2 & q_3 & q_4 \end{bmatrix}^T \tag{15}$$

The cost function is given by:

$$f(\boldsymbol{X}) = \frac{|K_2|}{min\{|K_1|, |K_3|, |K_4|\}} \tag{16}$$

The constraints can be described as:

$$g(\boldsymbol{X}) = K_3 + K_4 = 0 \tag{17}$$

$$\boldsymbol{X}_L \leq \boldsymbol{X} \leq \boldsymbol{X}_U \tag{18}$$

where $\boldsymbol{X}_U$ and $\boldsymbol{X}_L$ denote the upper and lower boundaries of weight coefficients. An appropriate optimization algorithm can lead to optimal design variables $\boldsymbol{X} = \begin{bmatrix} q_1 & q_2 & q_3 & q_4 \end{bmatrix}^T$, resulting finite LQR control. It should be noted that since LQR control is a control method in infinite domain, the optimal control gain is constant when the nominal parameters of the vehicle suspension are unchanged. Thus, only one optimization is needed, and it is not necessary to utilize the time-consuming optimization affecting the real-time performance of the control system.

### 3.3. Finite State LQR Control Law

After the optimal feedback gain is obtained, the control input at current moment can be given by:

$$u(k+1) = -\Big(K_{f1} x_1(k) + K_{f2} x_2(k) + K_{f3} x_3(k) + K_{f4} x_4(k)\Big) \tag{19}$$

According to Equation (1), the acceleration of sprung mass at the last moment is:

$$m_s \dot{x}_3(k) = -k_s x_1(k) - c_s(x_3(k) - x_4(k)) + u(k) \tag{20}$$

Considering $K_{f3} = -K_{f4}$, the current control input can be expressed as:

$$\begin{aligned} u(k+1) &= -\Big(K_{f1} x_1(k) + K_{f2} x_2(k) + K_{f3}(x_3(k) - x_4(k))\Big) \\ &= -\Big(K_{f1} x_1(k) + K_{f2} x_2(k) + K_{f3} \tfrac{u(k) - m_s \dot{x}_3(k) - k_s x_1(k)}{c_s}\Big) \\ &= -\Big(\Big(K_{f1} - K_{f3} \tfrac{k_s}{c_s}\Big) x_1(k) + K_{f2} x_2(k) + K_{f3}\Big(\tfrac{u(k)}{c_s} - \tfrac{m_s \dot{x}_3(k)}{c_s}\Big)\Big) \end{aligned} \tag{21}$$

As $K_{f2}$ approaches 0, the Equation (21) can be expressed as:

$$u(k+1) = -\left(\left(K_{f1} - K_{f3}\frac{k_s}{c_s}\right)x_1(k) + K_{f3}\left(\frac{u(k)}{c_s} - \frac{m_s\dot{x}_3(k)}{c_s}\right)\right) \tag{22}$$

The control input at the next moment can be expressed using the existing sensor outputs and actuator outputs, without estimating the certain states of the system. In this way, LQR control under finite states (or sensor outputs) is achieved, which greatly simplifies the design of control system.

In fact, there exists a great deal of control methods that aim to obtain the expression similar to Equation (22) through a large amount of data training. To avoid data training, the proposed FSLQR control adopts the optimization strategy for the desired control law through weakening the influence of unknown states. The unknown states are decided according to the output conditions of sensors. By this means, the proposed method realizes linear quadratic optimal control effect under finite sensor arrangement, which indicates great universality and practicality.

State estimation could take up the computing memories of the controller and lead to a cumulative effect of errors from the state observer to the controller module. It can be observed from Equation (22) that the intermediate state estimation is omitted, which enhances the practicability and reliability of the whole control system.

## 4. Examples Adopting Finite LQR Control

In this section, the actual control performance of FSLQR is studied. The suspension system shares the same dynamic parameters (as shown in Table 1) according to literature [21,22], and the uncertainty of the suspension system parameters is considered. Considering the accuracy of sensors, noise is added into the outputs of sensors accordingly [23]. To characterize the uncertainty of suspension parameters, the sprung and unsprung mass parameters of the actual suspension system are assumed to be uniformly and randomly distributed within a given range. For the convenience of performance evaluation, the simulations of systems with full state LQR control, FSLQR control, passive control, and PID control are conducted under impact and random road excitation. Figure 2 shows the Simulink model developed for quarter-car suspension model, which contains four control strategies: full state LQR control, FSLQR control, passive control, and PID control. Table 2 shows the parameters of full state LQR controller, FSLQR controller, and PID controller.

**Table 1.** The dynamics parameters of the suspension system.

| Parameter | Symbol | Value | Unit |
|---|---|---|---|
| Unsprung mass | $m_u$ | 110~118 | kg |
| Sprung mass | $m_s$ | 950~974 | kg |
| Tire's stiffness | $k_t$ | 101,115 | N/m |
| Tire's damping | $c_t$ | 14.6 | N·s/m |
| Suspension's stiffness | $k_s$ | 42,720 | N/m |
| Suspension's damping | $c_s$ | 1095 | N·s/m |
| Maximum travel | $z_{max}$ | 100 | mm |

**Table 2.** The parameters of the full state LQR controller.

| Parameter | Symbol | | | | Value | Unit |
|---|---|---|---|---|---|---|
| LQR weight coefficients | $q_1$ | $q_2$ | $q_3$ | $q_4$ | 2, 1, 5, 0 | — |
| FSLQR weight coefficients | $q_{f1}$ | $q_{f2}$ | $q_{f3}$ | $q_{f4}$ | 1, 9990, 2197, 0 | — |
| PID controller | $K_p$, $K_i$, $K_d$ | | | | 0, 80,000, 0 | N·s/m |
| Time step | h | | | | 0.001 | s |

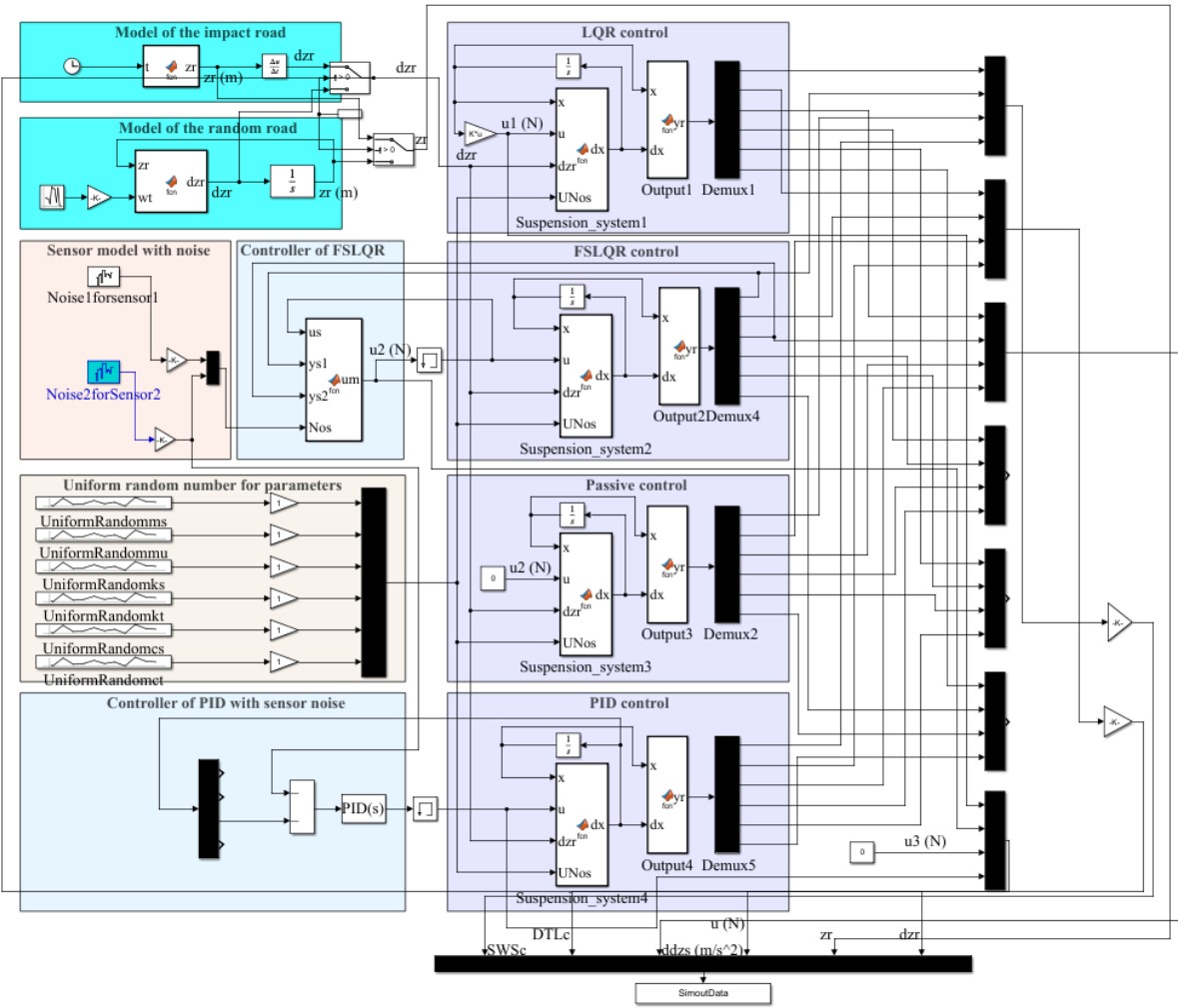

**Figure 2.** Quarter car suspension model.

### 4.1. Impact Excitation

The impact excitation of road is given by [22]:

$$z_r = \begin{cases} \frac{A_b}{2}\left(1 - cos\left(\frac{2\pi V}{L}t\right)\right), & 0 \leq t \leq \frac{L}{V} \\ 0, & t > \frac{L}{V} \end{cases} \tag{23}$$

where, $A_b$ and $L$ denote the height and length of impact excitation. $V$ is the speed of vehicle. It is assumed in the example that $A_b = 50$ mm, $L = 6$ mm, and $V = 35$ km/h.

Figures 3–5 depict the suspension dynamic travel coefficient responses of SWSc, the tire's loading coefficient responses of DTLc, and the sprung mass's acceleration responses under the impulse excitation, respectively. It can be observed from Figures 3 and 4 that SWSc and DTLc are both less than 1, which meet the requirements of vehicle pavement retention and suspension limit travel. When applying LQR control, FSLQR control, and PID control, SWSc and DTLc are much smaller than the passive control results, except SWSc is greater than the results of passive control at the beginning. Figure 5 shows that the response of spring mass acceleration under LQR control, FSLQR control, and PID control is much smaller than the results of passive control. The results indicate that the suspension system applying active control significantly improves the ride comfort of the vehicle.

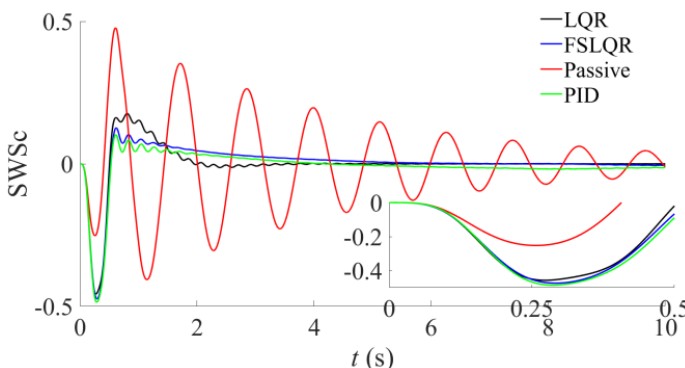

**Figure 3.** The suspension dynamic travel coefficient responses of SWSc under the impulse excitation.

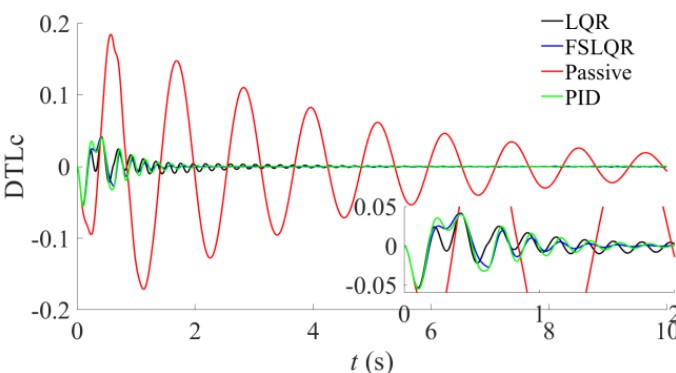

**Figure 4.** The tire's loading coefficient responses of DTLc under the impulse excitation.

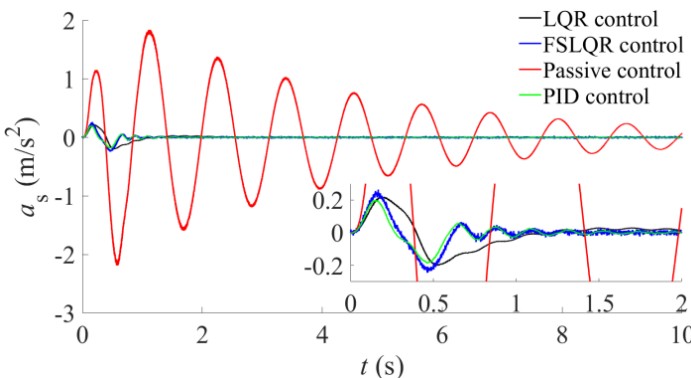

**Figure 5.** The sprung mass's acceleration responses under the impulse excitation.

From what has been discussed above, full state LQR control, FSLQR control, and PID control have similar response in values and are able to achieve desired control objectives. However, from the perspective of the smoothness of the response curve, full state LQR control and PID control with full state are slightly better than FSLQR control.

Full state LQR controller seems to promise better control performance. While, to ensure that performance, intermediate states such as suspension work stroke, tire work stroke, sprung mass acceleration, and unsprung mass acceleration need to be obtained in advance. Generally, the absolute speed of the sprung mass and the unsprung mass are difficult to obtain, and the acquisition of the tire work stroke is also challenging. As a result, full state LQR control of suspension system is limited. In addition, although PID control can improve vehicle comfort greatly, for the underactuated system like suspension, it is difficult for SISO method to consider other performance indexes. Inappropriate PID tuning can even lead to instability of other states.

Whereas FSLQR control satisfies the limitation of actual sensors and coordinates the suspension performance indexes. Without estimation of unknown states, FSLQR control simplifies the control system greatly and therefore enhances the practicability of the whole system.

*4.2. Random Excitation*

The filtered white noise excitation of road is given by [24]:

$$\dot{z}_r = -2\pi f_0 z_r + 2\pi n_0 \sqrt{G_q(n_0)V} * w(t) \tag{24}$$

where, $G_q(n_0)$ is pavement unevenness coefficient, and the value of that of Chinese national standard C-grade pavement is $256 \times 10^{-6}$ m$^3$. $f_0$ is the lower cut-off frequency and $f_0 = 0.0628$ Hz. $V$ is the velocity of vehicle. $w(t)$ is white Gaussian noise with power spectrum 1. $n_0$ is the frequency index, and $n_0 = 0.1$ m$^{-1}$.

Figure 6 shows the simulation curve of road pavement under the national standard C-grade and B-grade road surface with a vehicle speed of $V = 35$ km/h. Figures 7–9 depict the suspension dynamic travel coefficient responses of SWSc, the tire's loading coefficient responses of DTLc, and the sprung mass's acceleration responses under the random excitation, respectively. Table 3 displays the root mean square of responses for the convenience of analysis.

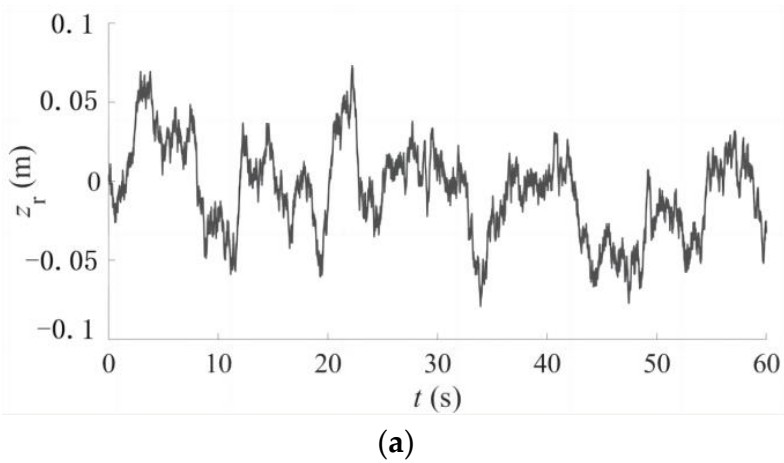

(**a**)

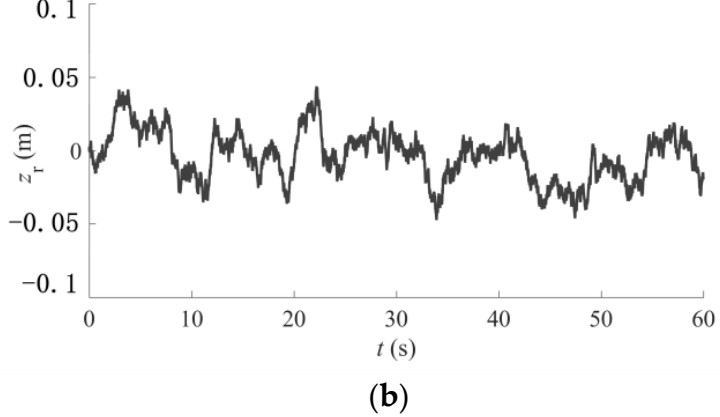

(**b**)

**Figure 6.** The simulation curve of road pavement. (**a**) Road pavement under the national standard C-grade road surface. (**b**) Road pavement under the national standard B-grade road surface.

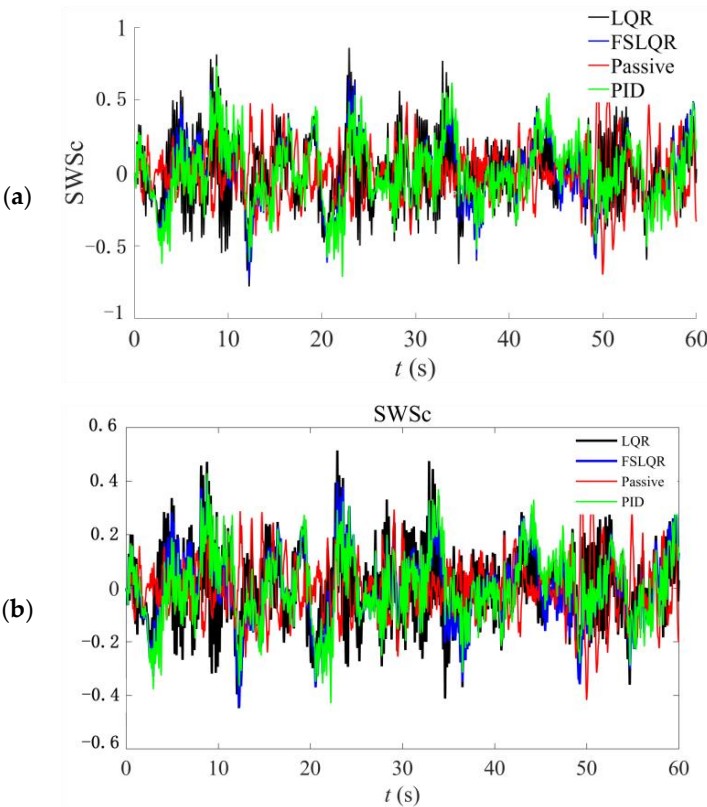

**Figure 7.** The suspension dynamic travel coefficient responses of SWSc under random excitation, (**a**) under the national standard C-grade road surface, (**b**) under the national standard B-grade road surface.

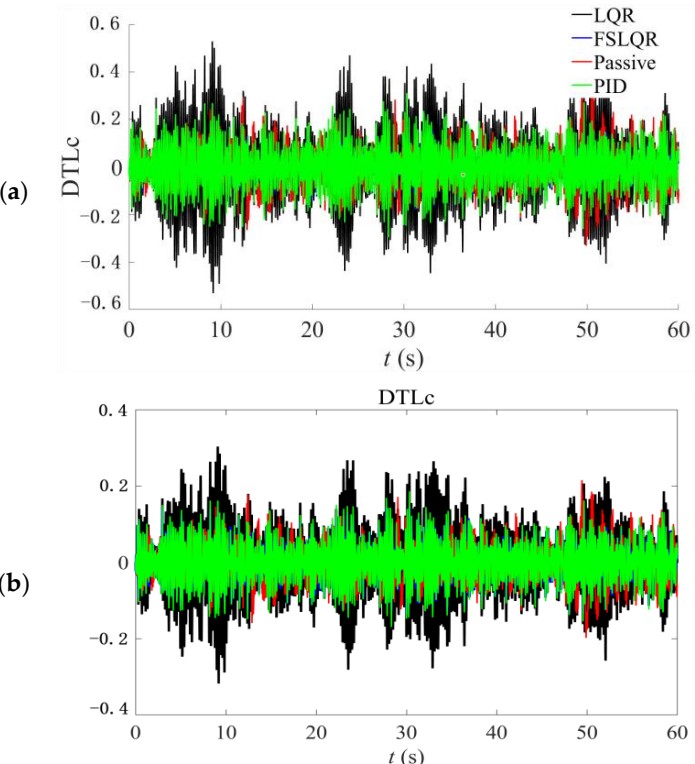

**Figure 8.** The tire's loading coefficient responses of DTLc under random excitation, (**a**) under the national standard C-grade road surface, (**b**) under the national standard B-grade road surface.

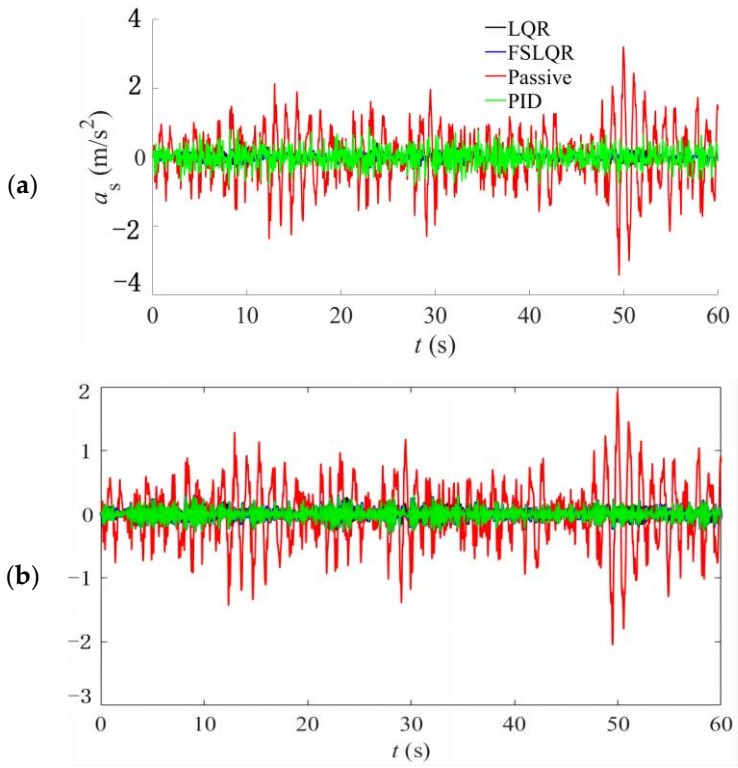

**Figure 9.** The sprung mass's acceleration responses under random excitation, (**a**) under the national standard C-grade road surface, (**b**) under the national standard B-grade road surface.

**Table 3.** The root mean square value of the random excitation response.

| Methods | Suspension Dynamic Travel Coefficient | | Tire's Dynamic Load Coefficient | | Sprung Mass's Acceleration (m/s²) | |
|---|---|---|---|---|---|---|
| Pavement level | C | B | C | B | C | B |
| LQR | 0.2285 | 0.1365 | 0.1537 | 0.0917 | 0.1364 | 0.0819 |
| FSLQR | 0.2240 | 0.1343 | 0.0708 | 0.0421 | 0.1531 | 0.0919 |
| Passive control | 0.1774 | 0.106 | 0.0846 | 0.0505 | 0.8065 | 0.482 |
| PID | 0.1416 | 0.1329 | 0.0531 | 0.0513 | 0.2532 | 0.0968 |

It can be observed from the results that the FSLQR control method produces more favorable results, whether using the B-grade or C-grade road surface. Figures 7 and 8 show that the SWSc and DTLc of B-grade and C-grade road surfaces are both less than 1, which meet the requirements of vehicle pavement retention and suspension limit travel. Although SWSc results of suspension under active control are obviously larger than that under passive control, the sprung mass acceleration under LQR and FSLQR control is much smaller. The results indicate that active control may increase the suspension travel to improve the comprehensive performance of vehicle suspension.

With passive suspension as the benchmark, according to Table 3, the performance index improvement of LQR control, FSLQR control, and PID control under random excitation of B-grade and C-grade road surfaces can be obtained, as shown in Table 4. The results of sprung mass acceleration and SWSc of LQR, FSLQR, and PID control also have little differences, with improvement around 80% and −26%, respectively. Numerically, full state LQR is slightly better than FSLQR and FSLQR is slightly better than PID control. However, both LQR control and PID control in full state deteriorate DTLCs, especially full state LQR control. Meanwhile, FSLQR improves DTLc well and has thus has the best comprehensive control performance. In addition, the effect of output noise is considered in

the actual sensor model in FSLQR, so the results of FSLQR control are more consistent with the actual situation.

**Table 4.** The improvement degree of different parameters' response performance under random excitation.

| Methods | Suspension Dynamic Travel Coefficient (%) | | Tire's Dynamic Load Coefficient (%) | | Sprung Mass's Acceleration (%) | |
|---|---|---|---|---|---|---|
| Pavement level | C | B | C | B | C | B |
| LQR | −28.76 | −28.77 | −81.47 | −81.58 | 83.00 | 83.01 |
| FSLQR | −26.04 | −26.70 | 16.66 | 16.64 | 81.00 | 80.93 |
| Passive control | 0.00 | 0.00 | 0.00 | 0.00 | 0.00 | 0.00 |
| PID | −25.81 | −25.37 | −1.54 | −1.58 | 79.89 | 79.92 |

In conclusion, FSLQR simplifies the design of control system by taking advantage of existing sensors. Meanwhile, FSLQR can obtain favorable comprehensive control performance and thus has strong practicability.

*4.3. Establishment of Finite State LQR Control System*

The FSLQR control can realize the active control of the suspension system by only using the limited sensor output and can also consider the performance indexes of the vehicle suspension system without the need to estimate the unknown states. By this means, FSLQR greatly simplifies the design of the control system. With the continuous debugging and modification of the FSLQR code, the stability and reliability of the code are becoming mature. To evaluate the performance of different control methods for impact and random road surfaces more conveniently, the "Vehicle Suspension FSLQR Control Simulation System" was developed based on the MATALB platform, as shown in Figure 10. To be more in line with the vehicular conditions, the system can also study the control performance of the vehicle suspension system under the uncertain suspension parameters through deviation at nominal parameters.

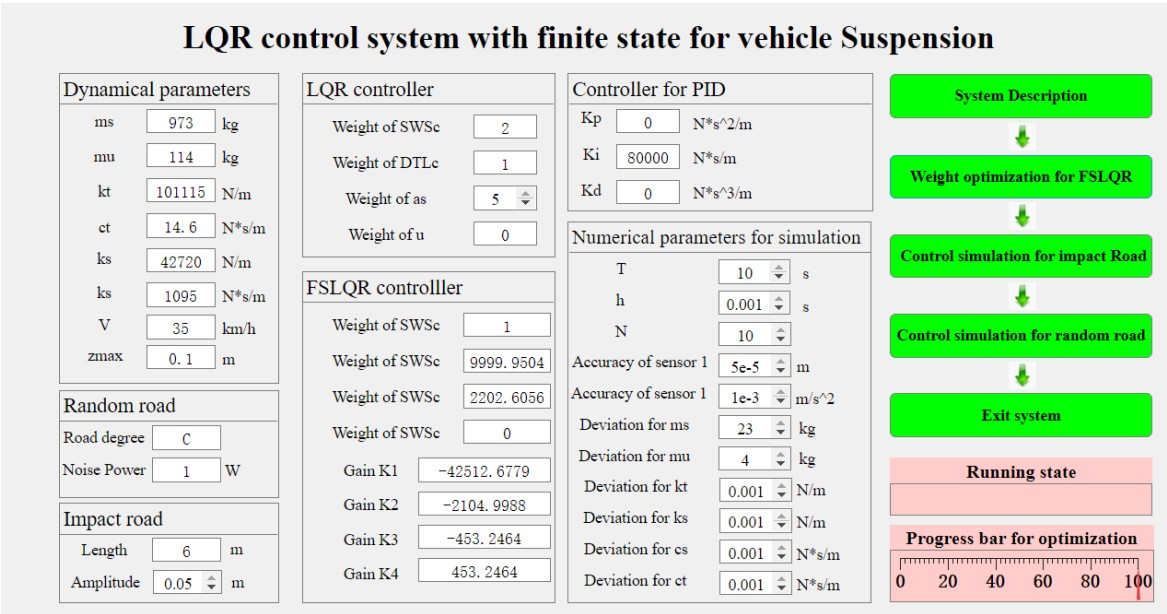

**Figure 10.** The establishment of LQR control system with finite state for vehicle suspension.

**5. Conclusions**

(1) Combining the linear quadratic regulator (LQR), finite sensor arrangement, and modern control theory, a finite state LQR control method is proposed for the application of suspension. Utilizing the information from finite sensors, an optimization model

  with LQR weight coefficients as design variables is established and linear quadratic optimistic control objective is achieved.

(2) Considering sensor noises and suspension uncertainties, the performance of the FSLQR method is evaluated through simulation comparison among four control methods under impact and random excitation. The results indicated that under impact excitation, full state LQR control, FSLQR control, and PID control have similar response values. However, full state LQR cannot achieve control objectives when the sensor arrangement is limited. Under random excitation, the ride comfort indexes are almost the same for full state LQR, FSLQR, and PID control. However, FSLQR improves DTLc greatly and the deterioration of SWSc is also small, indicating favorable comprehensive control performance.

(3) The proposed FSLQR overcomes the deficiency of the existing methods requiring intermediate states, and thus shows strong practicability. The FSLQR control method makes full use of the existing sensing information and does not need an estimation of unknown states. In this way, the design of the control system is greatly simplified, indicating strong practicability. Meanwhile, the proposed FSLQR control adopts the optimization strategy for the desired simple-formed control law, without massive training like a neural network algorithm. Thus, FSLQR has strong universality and is very suitable for the control system with finite sensing information.

(4) The "Vehicle Suspension FSLQR Control Simulation System" was developed based on MATALB for the evaluation of suspension systems with uncertainties in different control methods under impact and random excitation as well as suspension uncertainties.

**Author Contributions:** Q.F.: Conceptualization, Methodology, Software, Investigation, Formal analysis, Writing original draft. J.W.: Data curation, Writing—original draft. C.Y.: Visualization, Investigation. T.F.: Resources, Supervision. N.Z.: Software, Writing review & editing. J.Z.: Conceptualization, Funding acquisition, Resources, Supervision, Writing review & editing. All authors have read and agreed to the published version of the manuscript.

**Funding:** This research received no external funding.

**Institutional Review Board Statement:** Not applicable.

**Informed Consent Statement:** Not applicable.

**Data Availability Statement:** No new data were created or analyzed in this study. Data sharing is not applicable to this article.

**Acknowledgments:** The research is sponsored by the National Key Research and Development Program of China, China under Grants (No. 2019YFB2006402). The financial support is gratefully acknowledged.

**Conflicts of Interest:** The authors declare no conflict of interest.

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
