# Peer review of "Linear Quadratic Optimal Control with the Finite State for Suspension System"

_machines, doi:10.3390/machines11020127_

Round 1
Reviewer 1 Report
The paper deals with LQ optimal control with the finite state for suspension system. The main novelty of the paper is the avoidance of uncertain state variables in the control design, which only requires measurements by existing sensors of the vehicle. The paper is generally well written with some minor grammatical errors, for example:
· Suspension work stroke coefficient using finite state LQR control
· Finite state LQR tire improves dynamic tire load
· But these methods all result à However, these methods all result
· optimization is needed and there is no need to worry about à optimization is needed and it is not necessary to utilise
· Full state LQR controller seems promises à Full state LQR controller seems to promise.
In Section 3 describing the finite state LQR design, some clarification is needed for better understanding:
· Why is K3=-K4 desired in the design, please better explain.
· Why is the design variable vector is noted with X? It is confusing, since, since the states are noted with x.
My biggest concern with the presented method however, is that it assumes an active suspension, which is not as widespread as the semi-active suspension using MR damper. Authors should clarify, why active suspension was selected for the design.
Author Response
Please see the attachment, thank you.

Reviewer 2 Report
The article presents a complete and rigorous system simulation of an automotive damping system in which different control strategies of the active elements are compared with a Linear Quadratic Regulator. It is an extensive, serious, and well-presented study except for a few details that I will point out below:
1. The abstract is very badly written, I have not been able to find out the object of the article when reading it. I think it is important that the authors do not enter directly into this section without first indicating, in a few lines, what the objective of the article is.
2. I found the bibliography adequate, but I miss in the introduction a review of the types of damping currently used and what range of vehicles incorporate it. I have done it myself during the review and I was surprised to see that it does not seem to be exclusive to top-of-the-range vehicles. A few years ago our group modeled for BWM the adaptive anti-roll bar of the BMW 6-series and yet I have now been able to find active or semi-active dampers on vehicles of other brands. I think it is interesting to include this section, insofar as the greater the use of these systems the more important, or not, the contribution of the manuscript will be.
3. I have to admit that I find it difficult to accept as valid publications that do not contain experimental verification. I understand that for this type of study it is really complex and I do not think that authors must be required to do it, but it is necessary that they correct the following aspects:
A. Only one pavement profile has been used and at a single speed (Fig. 6), I understand that several should be analyzed and it should be verified that, in all of them, the new control method produces more favorable results.
B. As the comparison is made in terms of the control algorithm, it is also not possible to assess the real contribution of the article in technical terms. By this I mean that ignoring, as in my case, the control methods of existing systems and reviewing the bibliography, I see a coherent article but I do not know the degree of improvement that the authors' proposal represents in practical terms, beyond the use of fewer sensors. No comment on its possible practical implementation has been included.
As I have said before, I think the paper is serious and thorough, I thank the authors for their careful presentation but I think it needs a minor revision to address the above issues in order to be published.
Author Response
Please see the attachment, thank you.
